# The Control of Metabolic CO_2_ in Public Transport as a Strategy to Reduce the Transmission of Respiratory Infectious Diseases

**DOI:** 10.3390/ijerph19116605

**Published:** 2022-05-28

**Authors:** Marta Baselga, Juan J. Alba, Alberto J. Schuhmacher

**Affiliations:** 1Institute for Health Research Aragon (IIS Aragón), 50009 Zaragoza, Spain; mbaselga@iisaragon.es (M.B.); jjalba@unizar.es (J.J.A.); 2Department of Mechanical Engineering, University of Zaragoza, 50018 Zaragoza, Spain; 3Fundación Agencia Aragonesa para la Investigación y el Desarrollo (ARAID), 500018 Zaragoza, Spain

**Keywords:** airborne, CO_2_, collective transport, SARS-CoV-2, tramway, filtration, infectious diseases, epidemiology, public health, COVID-19

## Abstract

The global acceptance of the SARS-CoV-2 airborne transmission led to prevention measures based on quality control and air renewal. Among them, carbon dioxide (CO_2_) measurement has positioned itself as a cost-efficiency, reliable, and straightforward method to assess indoor air renewal indirectly. Through the control of CO_2_, it is possible to implement and validate the effectiveness of prevention measures to reduce the risk of contagion of respiratory diseases by aerosols. Thanks to the method scalability, CO_2_ measurement has become the gold standard for diagnosing air quality in shared spaces. Even though collective transport is considered one of the environments with the highest rate of COVID-19 propagation, little research has been done where the air inside vehicles is analyzed. This work explores the generation and accumulation of metabolic CO_2_ in a tramway (Zaragoza, Spain) operation. Importantly, we propose to use the indicator ppm/person as a basis for comparing environments under different conditions. Our study concludes with an experimental evaluation of the benefit of modifying some parameters of the Heating–Ventilation–Air conditioning (HVAC) system. The study of the particle retention efficiency of the implemented filters shows a poor air cleaning performance that, at present, can be counteracted by opening windows. Seeking a post-pandemic scenario, it will be crucial to seek strategies to improve air quality in public transport to prevent the transmission of infectious diseases.

## 1. Introduction

Public health strategies are modulated by adjusting to the development of knowledge about the transmission routes of COVID-19. The viral transmission of SARS-CoV-2 human–human has been described from direct respiratory dissemination and indirect dissemination. On the one hand, direct respiratory dissemination, where the symptomatic or asymptomatic patient expels contaminated particles in respiratory events, and, on the other hand, indirect dissemination or via fomites, where transmission is due to contact with contaminated surfaces. On the other hand, it is possible to differentiate between the droplet and bioaerosol models with indirect dissemination. While droplets predominate in close contact, bioaerosols can be transmitted through the air over time and distance [1]. Regarding this pandemic, the scientific community has redefined the concept of bioaerosol, extending its consideration to airborne particles smaller than 100 µm, based on evidence and common factors related to the aerodynamics of the particles [1,2]. The spread patterns of SARS-CoV-2 could not be explained by traditional epidemic models, where homogeneity in the transmission is assumed [3]. As recognized by the WHO in April 2021 [4], a predominance airborne way has been suggested compared to other propagation models [1,5].

The size of the SARS-CoV-2 virion varies between 70 and 90 nanometers [6,7], and an average concentration of the virus in the sputum of 7.0 × 10^6^ copies/mL and a maximum of 2.35 × 10^9^ copies/mL [8]. Consequently, the viral load occupies 2.14 × 106% del bioaerosol on average. With this value, Lee [9] estimated a theoretical minimum and initial aerosol size of 4.7 µm to contain SARS-CoV-2. However, experimental bioaerosol sampling studies suggest the presence of the virus in smaller particle sizes (even < 0.25 µm) [10,11,12,13]. Despite numerous factors influencing the airborne transmission of pathogens, such as dynamics or their aerial persistence, contagion events can only be explained by a medium and long-distance airborne transmission model [5]—for example, among small animals [14,15], from viral superspreading events [16], in the long-distance transmission where infected individuals do not come into contact direct [17], by asymptomatic individuals transmission rates [18], and by the prevalence of spread in closed spaces [19]. Specifically, a superspreading event affected public transportation. Shen et al. [20] reported a massive infection of 24/68 (35.3%) people from a single infected individual while being transported in a bus with air recirculation and poor ventilation.

At the pandemic’s beginning, this route of contagion was dismissed, and more attention was paid to contagion by droplets and fomites. Consequently, there was controversy about whether asymptomatic infected individuals could be transmitters of SARS-CoV-2. However, currently, it has been estimated that 44% (CI95; 30–57%) of secondary cases were infected during the incubation period [21], where the individuals were asymptomatic. The global acceptance of the COVID-19 airborne spread allowed an improvement in the preventive methods, including new techniques for epidemiological management, such as the measurement of exhaled carbon dioxide (CO_2_) as an indicator of the risk of contagion [2,22].

Carbon dioxide measurement began to be used in the 19th century to design ventilation systems in architecture [23]. In the pre-pandemic period, CO_2_ measurement helped improve academic performance in schools and colleges [24] and, sporadically, control infectious diseases [25]. Due to the COVID-19 pandemic, CO_2_ measurement has become one of the preferred preventive strategies to reduce the risk of contagion by aerosols [22,26,27]. The direct measurement of aerosols to determine the risk of contagion by SARS-CoV-2 is highly complex and expensive since it requires highly specialized equipment. While there are handheld instruments or simple sensors to direct measure of aerosol concentrations, these instruments present different limitations such as they can not discriminate human-exhaled versus environmental aerosols; usually, they cover a limited range of particle diameter and hardly measure the submicronic particles. To overcome these hurdles, the CO_2_ level has been suggested as an indirect indicator of respiratory infectious diseases’ transmission [22]. CO_2_ is co-expired with bioaerosols that may contain SARS-CoV-2 in infected people [28,29,30]. Its quantification provides an idea of indoor air renewal and establishes the risk of infection as it depends on the viral load [31]. Consequently, the measurement of indoor CO_2_ is suggested as a reasonable ventilation proxy for respiratory infectious disease. Through its reading, it is possible to determine what percentage of the air has been exhaled by another individual (y) according to the expression $y=C_{e}\;x+C_{a}\left(1-x\right)$, where $C_{e}$ corresponds to the concentration of CO_2_ in exhaled air (estimated at 40,000 ppm), $C_{a}$ to ambient CO_2_ concentration, and x to the fraction of exhaled air. For example, if we assume a basal value is 440 ppm (fresh air outdoors), a group of people manages to increase it to 2300 ppm. In that case, the approximate percentage of air that those individuals have already breathed will be around 4.7%.

Despite the ventilation rates being known to influence the concentration of microorganisms in the environment [32], the increase in the exhalation rate of aerosols depending on CO_2_ has been poorly explored [30]. The concentration of airborne particles and the level of CO_2_ cannot be directly related due to a disparity between the bioaerosols generated and the respiratory activity [28]. For example, aerosol generation during forced vocalization or coughing is not comparable to emission rates during respiration [33]. Thus, two different scenarios (for example, a library versus a gym) with similar CO_2_ levels have to be interpreted individually.

To date, COVID-19 superspreading events have been reported indoors [34,35,36,37,38,39,40]. Thus, the use of air renewal proxy indoors is crucial for maintaining spaces with a low risk of contagion by aerosols. Many countries are making high economic investments to equip schools with CO_2_ meters [41,42]. In addition, other isolated initiatives have successfully implemented this methodology in shopping centers [43], collective transport [44,45,46], offices [47], or university and school classrooms [47,48,49,50,51,52]. Specifically, a recent study in Italy reported an 82% reduction in secondary COVID-19 infections in schools where they controlled air renewal from CO_2_ measurements [53].

Currently, CO_2_ concentration limits have been proposed as a reference to minimize COVID-19 spreading. Usually, it is set between 700 and 1000 ppm regardless of the event [54,55]. Urban collective transport is one of the policies designed to promote sustainable cities [56]. To prevent respiratory infectious diseases spread, it is important to analyze the risk involved in every specific means of transport. References on the emission of bioaerosols in collective transport are scarce despite being the environment with the second-highest transmission rate of SARS-CoV-2. Lan et al. [57] point to 18% of cases in the transport sector, only behind the health sector (22%) in the transmission rate of COVID-19 disease. Before the pandemic, some reports pointed to metabolic CO_2_ concentrations of up to 3700 ppm in buses [58,59,60,61], suggesting poor air renewal. However, due to the pandemic, it has been possible to reduce it to <800 ppm by implementing simple ventilation measures [45]. The operating conditions of the subways require reinforcement of artificial ventilation, for which values close to 1000 ppm have been found [44,62,63].

Trams have similar characteristics to buses since they circulate outside, and the contribution of natural ventilation can substantially favor air renewal. However, no specific information on air quality in trams has been reported. This work evaluates the accumulation of metabolic CO_2_ in the Zaragoza Tram (Spain) in circulation under different conditions. On the one hand, the objective was to analyze the concentration of CO_2_ in different events (e.g., weekend versus midweek, with and without air recirculation or with different weather conditions). To compare air renewal regardless of the event, the ppm/person indicator was used. However, secondary air purification methods that affect contagion risk, such as added air filtration, must also be considered. Then, the performance of the installed filtration system is analyzed against the concentration of submicron aerosols (such as the airborne virus SARS-CoV-2). The work concludes with suggestions for measuring CO_2_ and recommendations to reduce the risk of contagion in collective transport.

## 2. Materials and Methods

### 2.1. Measurement of Metabolic CO_2_

The metabolic CO_2_ level was measured using Aranet 4 Pro meters (Aranet Wireless Solutions España SL, Madrid, Spain), with technical characteristics shown in Table 1. The increase in CO_2_ ($\Delta {\mathrm{CO}}_{2}$) was determined according to Equation (1):(1)$$\Delta {\mathrm{CO}}_{2}={\mathrm{CO}}_{2,\mathrm{indoors}}-{\mathrm{CO}}_{2,\mathrm{outdoors}}\;$$

The Urbos 3 tram models (CAF, Beasain ES) have a total length of 33 m, a width of 2.65 m, and a height of 3.2 m. They have a capacity of 200 seats, of which 54 are seats. Travelers wore a mask at all times, and the windows remained partially open during all routes. Eight CO_2_ meters were installed at a 2.25 m height at different points of the Tram, according to the distribution of Figure 1a. The objective was to obtain realistic and uniform measurements, representative of the level of exposure experienced by an average user without running the risk that the measurement would be altered due to the direct exhalation of the passengers. As shown in Figure 1b, the meters were installed on grab bars, for which it was necessary to manufacture anti-vandal housings with holes to guarantee air transfer.

### 2.2. Probability of Contagion Determination by the CO_2_ Level

The CO_2_ measurement was used as a tool to determine the risk of contagion. This was possible thanks to the theoretical model updated by Peng and Jiménez [22] and the Aireamos consortium [31].

The risk of airborne indoors transmission (for one person in one hour) P was described from an alternative equation to that of Wells–Riley [64] (Equation (2)), enunciated by Rudnick et al. [65] (Equation (3)): (2)$$P=1-e^{-n}$$
(3)$$P=1-\exp\left(\frac{Itqf}{n}\right)R$$
where I is the number of infected people in a space, t is the exposure time measured in hours, q is the number of pathogens spread per hour, f is the fraction re-inhaled $\left(\left[C-C_{0}\right]/C_{a}\right)$, n is the number of people exposed to the infectious individual, and R is the particle retention efficiency, that is, the fraction of retained aerosols by the PPE from the exposed individual. In turn, C is the concentration of CO_2_ indoors, $C_{0}$ outside, and $C_{a}$ the concentration exhaled during respiration, defined in parts per million (ppm). The value of *n* corresponds to the infectious dose inhaled by a susceptible person. However, Rudnick et al. [65] assumed some conditions for the model’s description: (1) the indoor air is thoroughly mixed, so the infectious aerosol generated can be found anywhere in the space. (2) The external concentration of CO_2_ remains constant during the event. (3) Removal of viral aerosols due to virus survival, filtration, or other mechanisms is negligible compared to ventilation.

Peng and Jiménez [22] applied another alternative to the Wells–Riley formulation regarding the COVID-19 pandemic. The authors derived analytical expressions for the probability of infection indoors through the concentration of CO_2_. The expected value of 〈n〉 can be calculated for an uninfected person, assuming the probability that the individual is immune $\eta_{in}$ according to Equation (4):(4)$$\langle n\rangle =\left(1-\eta_{in}\right)C_{p}BD\left(1-m_{in}\right)$$
where $C_{p}$ corresponds to the average number of viruses (quantos.m^3^), B to the respiratory rate of the susceptible person (m^3^ h^−1^) that will vary depending on the activity carried out at that time, D the event duration (h), and $m_{in}$ the filtration efficiency of the mask during inhalation. Consequently, assuming no pre-existence of viral aerosols before the event, the analytical expression for the expected value of $C_{p}$ can be described by Equation (5):(5)$$C_{p}=\frac{\eta_{in}\left(N-1\right)E_{p}\left(1-m_{ex}\right)}{V}\left(\frac{1}{\lambda }-\frac{1-e^{-\lambda D}}{\lambda^{2}D}\right)$$
where N is the number of occupants, $E_{p}$ is the exhalation rate of SARS-CoV-2 per infected person (quantos.h^−1^), $m_{ex}$ is the filtration efficiency of the mask during exhalation, V is the volume of air in the space (m^3^), and λ the global rate constant of virus infectivity loss (h^−1^), including all those mechanisms that may affect virus survival (filtration, ventilation, etc.). Assuming that the increase in CO_2_ (∆CO_2_) of the indoor air concerning that of the outdoor air is only produced by human activity, it can be described as follows (Equation (6)):(6)$$n_{\Delta CO2}=\Delta C_{p.CO2}BD\;$$
where the CO_2_ increment volume and the CO_2_ exhalation rate per person mixing ratio (∆CO_2_), in m^3^.h^−1^, can be described as (Equation (7)), where $\lambda_{0}$ corresponds at ventilation rate (h^−1^):(7)$$\Delta C_{p.CO2}=\frac{NE_{p.CO2}}{V}\left(\frac{1}{\lambda_{0}}-\frac{1-e^{-\lambda_{0}D}}{\lambda_{0}{}^{2}D}\right)$$

As a result of this model, Peng and Jiménez [22] propose an acceptable probability of infection limit of p=0.01%. Although it does not imply safety in any situation, since with high N and/or D and/or the event occurs many times, the probability of infection for the susceptible person is understated.

### 2.3. Studied Routes of the Zaragoza Tram

Eighty-eight round trips (44 complete trips) with an average of ~40 min each were analyzed. As shown in Figure 2, each complete path stops at 42 stations. The routes included in stations #7–#10/#33–#36, and #10–#16/#27–#33 correspond to the university area and the city center, respectively.

The CO_2_ meters were installed for three months in the vehicle. We hypothesized that the variation in the meteorological data obtained during the study days could translate into variations in the Tram’s ventilation capacity. The variables of interest for five reference days of December 2020 are shown in Table 2. According to data provided by the Zaragoza weather station, the average wind speed value in December was 3.25 (±1.65) m/s, so it can be considered that on days B, C, and D, the values of wind speed and maximum gusts were low. Low wind speed was an unfavorable condition for the natural ventilation of the Tram.

### 2.4. Determination of Filtration Efficiency against Submicron Particles and Filters’ Pressure Drop

The filter’s performance was studied in-vitro to assess its effectiveness against submicron particle sizes, as is the case with the SARS-CoV-2 virus and other respiratory viruses. The filter used during the tests was specially implemented due to the current COVID-19 pandemic (Coarse 75% according to UNE-EN ISO 16890, Merak Long Life Filter, Madrid SP). As depicted in Figure 1, two filters were arranged in two HVAC units installed in the Tram, which drive a total flow of 2800–3300 m^3^/h, with an air ratio of 1:3 fresh/return air.

As shown in Figure 3a, NaCl aerosols were produced using a Topas-ATM226 generator with a saline solution of sodium chloride (3 wt.% NaCl in distilled water). Microdroplets were evaporated using a tubular silica air dryer to produce solid particles. The particle size distribution (Figure 3b) inside the cabin was measured using an SMPS TSI 3936 composed of an electrostatic classifier (DMA TSI 3081) and a condensation particle counter (CPC TSI 3782). An 0.6 L/min flow rate drags the particles. The filter was placed between bronze discs sealed with Teflon tape, with 30 × 20 mm Teflon washers on each side. The desired flow rate was adjusted variating the exposed filter area (2.05, 4.1, and 8.1 mm). The measurements lasted 120 s and were made in duplicate. Measurements were made passing through a free tube between measurements to calculate relative efficiency according to Equation (8), where $C_{up}$ stands for concentration upstream and $C_{down}$ stands for concentration downstream. The retention efficiency is expressed in global efficiency as ‘global number of particles’:(8)$$\eta =100\times \frac{C_{up}-C_{down}}{C_{up}}\;$$

According to Bernoulli’s principle, the pressure drop was carried out using alcohol columns connected to the free ends of the tubes. Measurements were also made with a 0.6 L/min volumetric flow rate.

### 2.5. Statistic Analysis

The statistical analysis of the data has been carried out using the R-UCA v.4.0.2 software (University of Cadiz Spain, 2017) [66]. Mean comparisons were made with the Student’s *t*-test at a 99% confidence interval (CI99).

## 3. Results

### 3.1. CO_2_ Levels along the Route Are Closely Related to Occupancy

As shown in Figure 4a,b, the increase in the CO_2_ concentration inside the Tram gradually increases as it approaches the city’s downtown area #33–#36 and #27–#33 stations; approximately, at minutes 20 and 70 on the outward and return routes, respectively. The CO_2_ increase corresponds to the difference between the Tram indoor values concerning the external reference value (atmospheric) registered with sensor #8. Analyzing the increment makes it possible to determine the global CO_2_ concentration corresponding to metabolic CO_2_ to rule out possible external contamination. The calculated ppm/person ratio (Figure 4c) suggests a concentration of ∆CO_2_ in the final areas of the route associated with an accumulation of CO_2_ in the vehicle, which begins to be evident after driving through the city center. It may be explained because the number of travelers increases in the city center and accumulates CO_2_ not recirculated at subsequent stops. On average, the Tram doors open for 16.6 ± 3.6 s at each stop.

∆CO_2_ concentration is closely related to tram occupancy (Figure 5a), although there is some dispersion associated with external variables (Figure 5b). In absolute CO_2_ values, the maximum average was 835 ± 232 ppm, reaching a maximum value of 1229 ppm. In contrast, the lowest average was 541 ± 82 ppm. The pattern of ∆CO_2_ concentration on weekdays compared to weekend days is different, although it follows similar trends. As shown in Appendix A, the average of the trips made on weekends in the morning was 565 ± 318 ppm; in the afternoon, it was 580 ± 323 ppm, and, at night, it was 602 ± 330 ppm. On weekdays, an average of 592 ± 319 ppm was obtained in the morning, 595 ± 324 ppm in the afternoon, and 541 ± 292 ppm at night.

### 3.2. CO_2_ Levels Distribution Is Similar at Different Points Inside the Tram

∆CO_2_ dispersion measurements at the different points of the Tram were assessed using the records from each sensor, as shown in Figure 6. Passenger occupancy is rarely uniform along the Tram, and differences in the capacity distribution can lead to spatially disparate values. The average Relative Standard Deviation (RSD) was determined to determine the homogeneity of the CO_2_ distribution in the Tram. The RSD of 0.09 ± 0.02 suggested that the measurements were relatively homogeneous, although accumulation tendencies are typically observed in the central area of the Tram.

### 3.3. Improving the Air Renewal by the Closing of the Air Return

To study the influence of the return of air from inside the Tram to the air conditioning equipment, we worked with the data obtained through Sensors #3 and #7, located just below the grilles of the air return ducts. Days C and D were selected as a reference for the study due to the similarity between meteorological variables. The ∆CO_2_ varies when the air return is closed, as deduced in Figure 7. From the analyses carried out, the extreme values at the beginning and end of the route corresponding to the accumulation of gas in the Tram have been removed, offering a more realistic view of the internal atmosphere during the tour. Under these conditions, the average ppm/person rate without return was 3.5 ± 0.1 (Sensor #3) and 5.1 ± 0.1 ppm (Sensor #7) without air return, and 4.9 ± 0.7 (Sensor #3) and 6.0 ± 0.4 (Sensor #7) with air return. A reduction in ∆CO_2_ between 9% and 36% can be seen concerning air return ∆CO_2_ levels.

### 3.4. Wind Speed Contributes to Increasing Ventilation Rates

The days were divided depending on the wind speed into Set A (Day A and E) and Set B (Day B to D). Sets were made to assess whether the weather plays a crucial role in the ventilation. The objective of this section is to compare the ventilation pattern on days with different weather, with special attention to the average wind and gusts and the average temperature. The average weather conditions of interest for the days of each Set are shown in Table 3. In Set A, the average wind speed was 3.9 ± 1.1 m/s, while, in Set B, it was 1.8 ± 0.1 m/s, with maximum gusts of 10.0 ± 1.6 m/s and 5.6 ± 0.6 m/s, respectively. Figure 8a,b represent the ppm/person index for Set A and Set B, respectively. In addition, 88.4% of the ppm/person indices was higher in Set B than Set A. It was found that the means of the data from Set B were significantly lower than those from Set A using a hypothesis contrast (CI99; −3.26–−1.38). Limits are harmful in the CI, confirming that higher data on Set A. A Student’s *t*-test shows that the average ∆CO_2_ concentration increases on days with lower wind speeds are higher. However, the difference between the ppm/person index in Set B compared to Set A is 2.3 ± 3.3 ppm, compared to averages of 5.2 ± 3.7 ppm (Set A) and 7.5 ± 3.0 ppm (Set B), which represents a reduction of between 31 and 44%. These data suggest that the weather can substantially affect the recirculation of air inside the Tram, as shown in Figure 8.

### 3.5. Tram Speed Does Not Influence the Indoor Ventilation Rate

Tram speed while circulating did not seem to have a substantial effect on the reduction of ∆CO_2_ (Figure 9a), nor on the average reduction rate of ∆CO_2_ at 3 min (Figure 9b). The Student’s *t*-test showed no significant relationship between the rate of reduction of ∆CO_2_ compared to two different speed ranges: 1–20 km/h and 21–40 km/h. It may be due to the flow of the HVAC system, which generates internal drafts so that the inflow of air through the window does not alter the ventilation rates substantially.

### 3.6. The Filtration System Is Not Efficient against Submicron Matter

The tests have been carried out with the filter usually installed on the Tram (Coarse 75% Filter Media). However, the Coarse 75% filter specially implemented due to the current COVID-19 pandemic has been characterized in the laboratory. The Air Changes per Hour (ACH) of the Zaragoza Tram remained in the unit’s regular operation at 25 ACH. As shown in Figure 10, the Coarse 75% filter presented an approximate retention efficiency of 27.9% for 300 nm particles at a flow rate of ~2500 m^3^/h. The filtering efficiency decreases up to 2.4 and 2.3% using ~162 and ~622 m^3^/h, respectively. The largest particles present more inertia at high flow rates [67,68], resulting in a higher retention rate in the filter medium. Even though the clogging of NaCl particles observed in the head loss tests may have overestimated these results, which could be seen as an increase up to 440 Pascals of pressure drop (Table 4), this flow would be the most representative of the working conditions in the HVAC of the Tram system.

### 3.7. Probability of Infection

The probability of infection and the attack rate were calculated following the models proposed by Peng and Jiménez [22] and Aireamos [31]. Based on the average daily CO_2_ values collected in Appendix A, an attack rate of 0.06% was determined in the least favorable case (higher CO_2_ values) and an attack rate of 0.04% in the average case (global average CO_2_). According to Peng et Jiménez’s proposal, the probability of contagion <0.01% is acceptable, so the Tramway did not represent a high risk of contagion under the conditions studied, as shown in Table 5.

## 4. Discussion

SARS-CoV-2 bioaerosols dissemination in infected individuals’ exhalation is widely demonstrated [69,70,71]. In addition, the virus’s presence and persistence in the environmental air have also been extensively studied [10,11,72,73,74,75,76,77,78]. Given the apparent predominance of the airborne route of transmission of COVID-19, various strategies have been investigated to mitigate the risk of contagion. Public transport environments represent the second sector with the highest transmission of SARS-CoV-2, only behind the health sector [57]. However, computational studies point to a 1.5–1.6% attack rate [79,80]. Even though numerous works have been aimed at evaluating the behavior of bioaerosols in collective transport by computational fluid dynamics [81,82,83,84,85], extrapolation to actual conditions is an enormous limitation. One of the strategies that allow the indoor ventilation rate to be quantified in situ is the measurement of CO_2_, which has positioned itself as a standard for air control [22,86,87].

In this work, CO_2_ measurements were collected in 88 round trips, which is equivalent to more than 79,200 records obtained from eight sensors strategically distributed in the Tram. The distribution of CO_2_ throughout the vehicle follows a similar trend (RSD 0.09 ± 0.02), so the location of the HVAC systems and natural air intakes seem to favor all points of the Tram equally. The average absolute CO_2_ of all the routes studied was 685 ± 59 ppm (572 ± 75 ppm–835 ± 232 ppm). This value suggests that the percentage of air already breathed is <0.7%. Considering the virus emission rates in exhaled breath [71,88], average time spent in the Tram (~7 min), and the mandatory use of facemasks, the interior of the vehicle does not represent a risk space of contagion by aerosols (probability of infection [22],  p=0.01%; attack rate < 0.1%) in the most unfavorable scenario (844 ppm average; 1571 ppm maximum). In this sense, Moreno et al. [78] reported attack rates between 0.00–0.72% in buses depending on the respiratory activity, bus air conditions, and the infected individual without a mask. Thus, the environment of the bus at that time was more dangerous.

Before the pandemic, some reports pointed to metabolic CO_2_ concentrations of up to 3700 ppm in buses [58,59,60,61]. However, a study on the bus in Barcelona (Spain) points to concentrations close to 1000 ppm that can be easily reduced to <800 ppm by implementing simple ventilation measures (i.e., opening windows) [45]. The operating subways conditions require a reinforcement of artificial ventilation, for which values close to 1000 ppm have been found [62,63].

In this paper, we propose using ppm/person indicator as a measure that allows ∆CO_2_ levels comparison on different days and circumstances. A key aspect and an obvious one is the increase in ∆CO_2_ as the number of passengers increases. Analyzing the ∆CO_2_ data measurements, a gradual increase in CO_2_ concentration could be misinterpreted as an accumulation. However, looking at the ppm/person ratio, it can be seen that the increase in ∆CO_2_ comes from an increase in capacity.

Trams are typically similar to buses since they circulate outside and substantially favor measures to reinforce natural ventilation. In this work, it was found that the speed of the external wind reduced the ppm/person rates to around 2.3 ± 3.3 ppm. Although it seems a slight benefit, it represents a reduction of between 31 and 44% compared to days with less wind. Closing the air return (total external air intake) favored ventilation, reducing the ∆CO_2_ level between 9 and 36%. Tram speed did not affect ventilation rate, at least in two data sets with different speed ranges (2–20 km/h versus 20–40 km/h). However, the data could not be compared with the stopped Tram since the conditions were different at that moment. There are no sources of CO_2_ generation (there are no passengers), and the doors open entirely, so the air is wholly recirculated in a few minutes.

Favoring natural ventilation (opening windows), the HVAC system, and the use of masks have been shown to significantly reduce the risk of transmission [30,79,84,89,90]. Masks reduce the bioaerosols emission variably, depending on the type of mask and the aerodynamics of the scattered particles [91,92,93,94]. In addition, HVAC systems should consider the filter, but it is also possible to optimize it to maintain adverse thermodynamic conditions for the virus [95,96].

One of the most significant limitations of CO_2_ measurement is that its interpretation cannot be generalized but must be individualized. Aerosol generation fluctuates substantially depending on the individual’s respiratory event [2,33,97,98,99]. In addition, environmental conditions directly influence the spread and persistence of the virus [2,100,101]. Therefore, it is not easy to define an effective viral load dependent on CO_2_, at least in absolute terms. However, this and other studies demonstrate the effectiveness of CO_2_ measurement to implement effective air renewal patterns and reduce the risk of transmission of infectious diseases.

## 5. Conclusions and Recommendations

This work suggests that the measurement of the ∆CO_2_ concentration inside collective transport constitutes a cost-efficiency strategy that can reduce the rates of spread of the respiratory virus by aerosols, as is the case of the SARS-CoV-2 virus. In this work, the interpretation of the exhaled CO_2_ levels per person (ppm/person) has made it possible to analyze the behavior of the air inside the Zaragoza Tram. Maintaining the typical parameters of the HVAC units and implementing the partial opening of the windows, the maximum CO_2_ level was 1249 ppm. On average, 835 ± 232 ppm have not been exceeded on any of the routes studied, which indicates that air recirculation is adequate for vehicle occupancy. In addition, the absolute CO_2_ in all the routes studied was 685 and 690 ± 59 ppm, on average and median, respectively. However, it must be considered that capacity was reduced on the studied days due to the COVID-19 pandemic restrictions. It presents a limitation when extrapolating the data to post-pandemic operating conditions. The passengers’ exposure to the Tram air must also be considered since the average route usually lasts around 7 min, and passengers wear a mask and keep their distance when possible. Under the conditions studied, the following recommendations are suggested to reduce the risk of infection by aerosols and/or improve ventilation performance:
Maximize outside air intake: by opening windows, increasing door openings in stations, and minimizing the rate of return air in HVAC units;Completely recirculate the air between outbound and return routes to avoid exposing new passengers to the air breathed by previous passengers;Consider implementing efficient filtration systems against particles (0.1–100 μm) instead of coarse-type filters, efficient against pollen or dust. Additionally to filtration systems, other air purification technologies can be beneficial in improving air quality. Even so, its performance needs to be demonstrated under operating conditions and not just in the laboratory or theoretically;Limit the respiratory activity of passengers to calm breathing and speech and the use of masks and other personal protection equipment and promote interpersonal distance.


In addition, from experience gathered during the CO_2_ measurement experiments in public transport, the following recommendations can be drawn:
Initially, characterizing the distribution of CO_2_ inside the vehicle is essential so that the location of the sensors allows representative measurements of the space to be taken;Analyzing the increase in CO_2_ instead of absolute CO_2_ allows for quantifying only the CO_2_ generated by passengers, discriminating external pollution. Additionally, we propose to use the ppm/person ratio as the main indicator to compare the exhaled CO_2_ measurements on different scenarios. This ratio can be easily calculated by dividing the increase by the number of people. For example, if the increase in CO_2_ is 500 and there are 50 people, the ratio will be 10 ppm/person. In case of studying two separate days, for example with different weather, we can find that one day the ratio is 10 ppm/person and another day it is 30 ppm/person. With this information, we can determine how the change of variables affects independently of the occupation.Place the gauges at a sufficient height to avoid the direct exhalation of the passengers. For example, they were placed 2.25 m above ground level for this work. Moreover, locating meters near doors and windows should avoid underestimating CO_2_ levels.Evaluate weather conditions, especially airspeed, to interpret the measurement results on measurement days correctly. For example, in our study, the weather substantially affected the ventilation ratio inside the Tram. On the days with the greatest wind, ppm/person rates of up to 44% lower were recorded with respect to the days with the least wind.Recording occupancy levels (number of passengers) is essential to estimate the ventilation rate and to be able to compare data in different samples.Deduct the minimum number of meters to obtain representative measurements of the space. The heterogeneity in vehicle occupancy requires a consistent distribution of meters. For example, a meter was placed for every 35 m^3^ of air in this work.Considering the respiratory activity of the vehicle occupants is desirable when normalizing the ppm/person rates. In addition, the CO_2_ records must be individually interpreted depending on variables such as interpersonal distance, the use of masks or other PPE, and the implemented filtration systems (or other air purification devices).


Under the conditions studied, the Tram does not present itself as a space with a high risk of infection by aerosols (by using Aireamos Covid Risk Airborne tool [31]; see Section 3.7). Air quality monitoring began to gain popularity due to the COVID-19 pandemic. However, once the focus is on the air [2], a post-pandemic scenario presents uncertainty when the windows are completely closed and the capacity increases. Consequently, it will be necessary to implement a standard that allows air quality to be regulated in these post-pandemic conditions. The poor filtration performance against the submicronic matter of the typically implemented filters is a significant limitation. It is necessary to find new air control and purification strategies that reduce the risk of disease transmission in the future.

## Figures and Tables

**Figure 1 ijerph-19-06605-f001:**
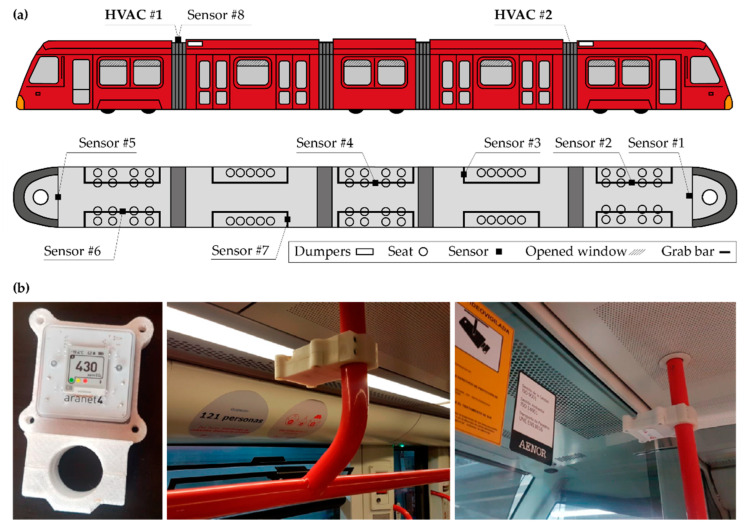
Schematic representation (**a**) of the meters distribution in the Tram and (**b**) installed sensors in the Tram. Where, # refers to the meter ID.

**Figure 2 ijerph-19-06605-f002:**
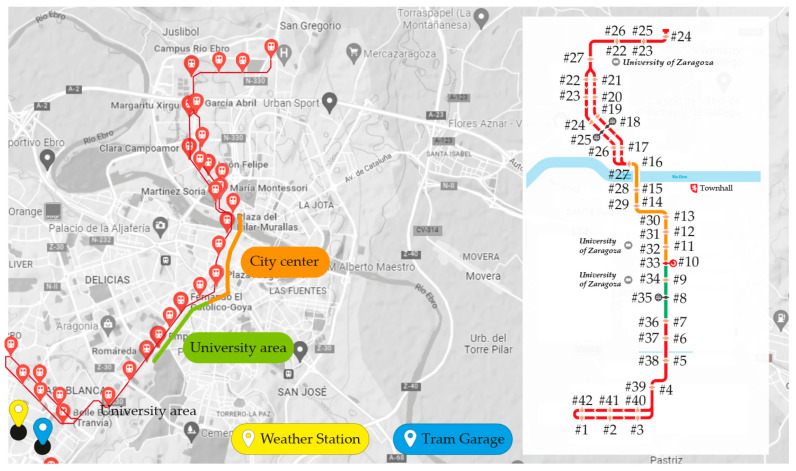
Route of the Zaragoza Tram. Where, # refers to the station ID.

**Figure 3 ijerph-19-06605-f003:**
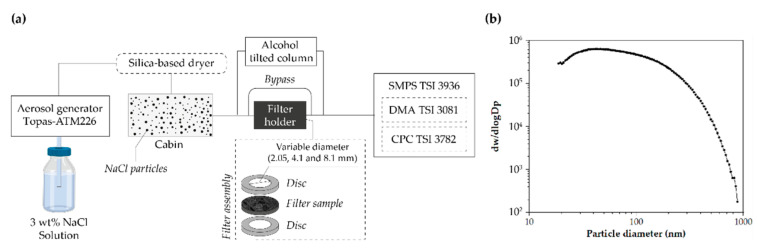
Performance test. (**a**) Diagram of the equipment used to characterize the filters and (**b**) particle concentration distribution for efficiency determination measurements in the range 0.1–1.0 μm.

**Figure 4 ijerph-19-06605-f004:**
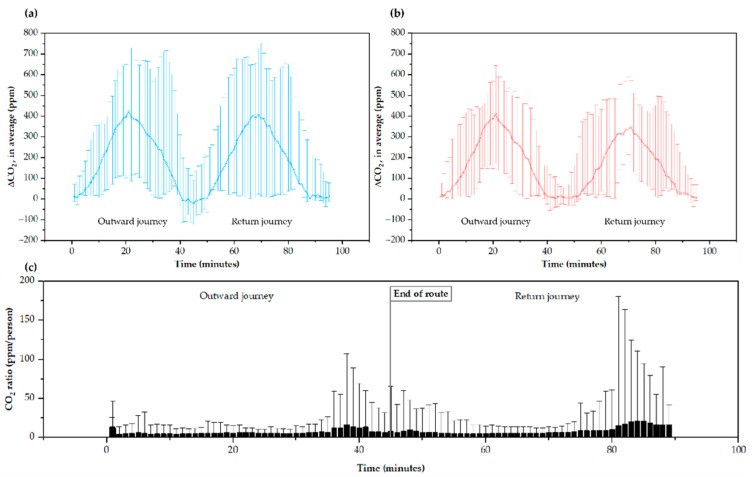
CO_2_ increment average levels (**a**) in all weekday and (**b**) in all weekend routes, and (**c**) ppm/person ratio average and maximum gap along routes. The error bars in (**a**,**b**) correspond to the difference between the maximum/minimum data and the average data of all studied routes.

**Figure 5 ijerph-19-06605-f005:**
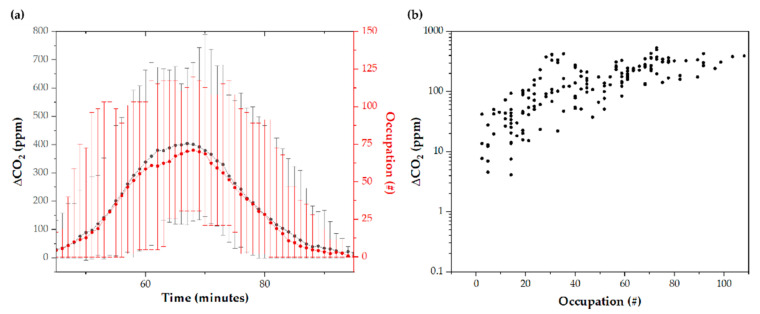
∆CO_2_ increment and tram occupancy as (**a**) a function of time, and as (**b**) a function of tram occupancy. The error bars correspond to the difference between the maximum/minimum data and the average data of all the studied routes.

**Figure 6 ijerph-19-06605-f006:**
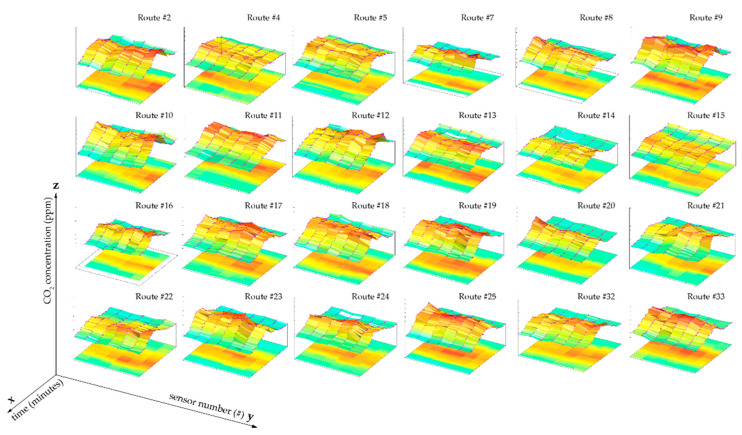
Distribution of ∆CO_2_ in the Tram (*z*-axis) as a function of time (*x*-axis) and ∆CO_2_ measures (*y*-axis) on routes #2, #4–#25, and #32–#33, where there is homogeneity in the CO_2_ measurement along the tram and a strong relationship with occupancy.

**Figure 7 ijerph-19-06605-f007:**
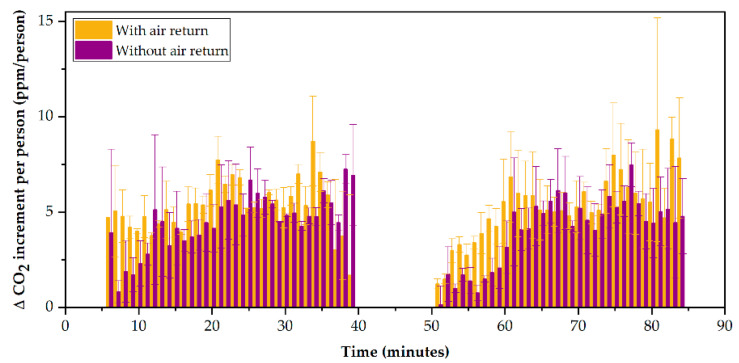
Registered ppm/person values depending on the air return.

**Figure 8 ijerph-19-06605-f008:**
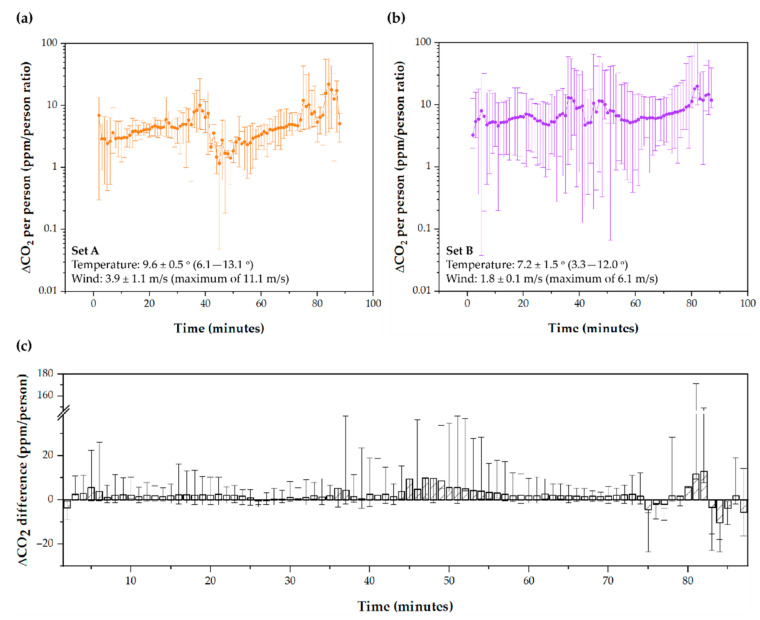
∆CO_2_ per person ratio (ppm/person) in (**a**) Set A, and in (**b**) Set B depending on time; (**c**) difference between Set B and Set A depending on time.

**Figure 9 ijerph-19-06605-f009:**
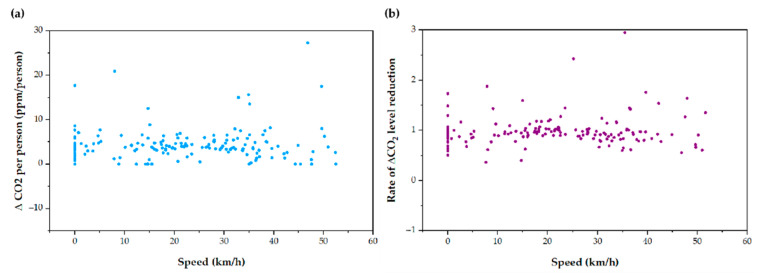
CO_2_ measurements depending on Tram speed. (**a**) ∆CO_2_ per person ratio, and (**b**) average reduction rate of ∆CO_2_ depending on the Tram speed.

**Figure 10 ijerph-19-06605-f010:**
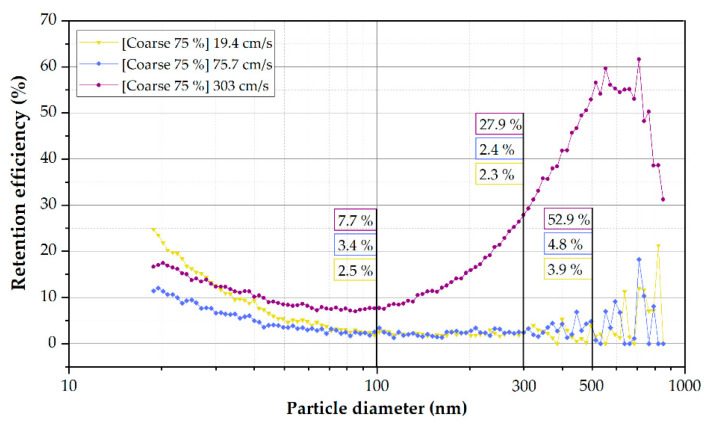
Coarse 75% filter retention efficiency depending on the particle diameter at different speeds (flow rates).

**Table 1 ijerph-19-06605-t001:** Technical characteristics of the Aranet 4 Pro meters.

Parameters measured	CO_2_	<9999 ppm
	Temperature	0–50 °C
	Relative humidity	0–85%
	Atmospheric pressure	0.3–1.1 atm
Sensor type	N-DIR (Non-Dispersive Infrarred)	
Communication technology	Bluetooth (−12–4 dBm)	
Sampling frequency	1 min	
Precision	±50 ppm (CO_2_)	
Dimensions/Weight	70 × 70 × 24 mm/104 g	

**Table 2 ijerph-19-06605-t002:** Meteorological variables of the reference days A, B, C, D, and E. Information prepared by the Agencia Estatal de Meteorología of Spain (data collected at the Valdespartera Station, Zaragoza Spain, 23 December 2020).

Day	$T_{aver}$	$T_{min}$	$T_{max}$	$D_{w}$	$W_{s,aver}$	$W_{s,max}$	$P_{max}$	$P_{min}$
A	9.6 °C	6.1 °C	13.1 °C	30°	3.1 m/s	8.9 m/s	996.8 atm	990.0 atm
B	8.2 °C	4.4 °C	11.9 °C	16°	1.7 m/s	6.1 m/s	996.8 atm	990.0 atm
C	5.4 °C	3.3 °C	7.4 °C	10°	1.9 m/s	5.0 m/s	998.2 atm	996.0 atm
D	7.9 °C	3.8 °C	12.0 °C	16°	1.9 m/s	5.6 m/s	994.7 atm	990.8 atm
E	8.9 °C	6.2 °C	11.6 °C	31°	4.7 m/s	11.1 m/s	994.9 atm	992.4 atm

$T_{aver}$: temperature (average); $T_{min}$: temperature (minimum); $T_{max}$: temperature (maximum); $D_{W}$: wind direction; $W_{s,aver}$: wind speed (average); $W_{s,max}$: wind speed (maximum); $P_{max}$: atmospheric pressure (maximum); and $P_{min}$: atmospheric pressure (minimum).

**Table 3 ijerph-19-06605-t003:** Meteorological variables of the reference Sets A and B. Information prepared by the Agencia Estatal de Meteorología of Spain (data collected at the Valdespartera Station, Zaragoza Spain, 23 December 2020).

Day	$T_{aver}$	$T_{min}$	$T_{max}$	$D_{w}$	$W_{s,aver}$	$W_{s,max}$
Set A	9.6 ± 0.5 °C	6.1 °C	13.1 °C	30 ± 0.5°	3.9 ± 1.1 m/s	11.1 m/s
Set B	7.2 ± 1.5 °C	3.3 °C	12.0 °C	14 ±3.5°	1.8 ± 0.1 m/s	6.1 m/s

$T_{aver}$: temperature (average); $T_{min}$: temperature (minimum); $T_{max}$: temperature (maximum); $D_{W}$: wind direction; $W_{s,aver}$: wind speed (average) and; $W_{s,max}$: wind speed (maximum).

**Table 4 ijerph-19-06605-t004:** Conditions used in the filtration tests and pressure drop determination.

Area	Flow Rate	Velocity in Filter	Pressure Drop
2281.6 cm^2^	~161.8 m^3^/h	19.4 cm/s	6 Pa
2281.6 cm^2^	~621.8 m^3^/h	75.7 cm/s	34 Pa
2281.6 cm^2^	~2488.8 m^3^/h	303.0 cm/s	440 Pa

**Table 5 ijerph-19-06605-t005:** Determination of the infection probability and attack rate by aerosols in two different scenarios simulates the Tram’s ventilation conditions, using Covid Risk Airborne [31] based on the model Wells–Riley [64].

Scenario	Facemask	Occupation	Exposure Time	CO_2_ Level	Variant	p	Attack Rate
#1 (global average)	Surgical mask	60 pax	10 min	689 ppm (average)/1038 ppm (max)	Omicron	0.01% *	0.04% *
#2 (maximum average)	Surgical mask	60 pax	10 min	810 ppm (average)/1520 ppm (max)	Omicron	0.01% *	0.06% *

* Considering a 78% vaccination with a proportionate immunity of 70%; a cumulative incidence (CI) of 1150 to 14 days/100,000 hab.

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
