# Peer review of "The Control of Metabolic CO2 in Public Transport as a Strategy to Reduce the Transmission of Respiratory Infectious Diseases"

_ijerph, 2022, doi:10.3390/ijerph19116605_

Round 1

Reviewer 1 Report

Overall, this work is interesting and important, I only have a few comments:

1 Lines 74-75, indeed, it is not that difficult to measure the aerosol directly by using the handheld instruments or simple sensors.

2 Please provide reference for this sentence “CO2 is co-expired with bioaerosols 77 that may contain SARS-CoV-2 in infected people.”

3 Figure 1, whether there exists any vent at the top of the Tram?

4 In the tram, whether the air is fully well mixed?

5 Lines 208-210, the authors need to mark out the location of the meteorological station in Figure 2, if the meteorological station is far away from the tram station, I do not think the data can represent the ventilation capacity.

6 All the figures presented in this work are not that clear, especially Figure 4 and Figure 6, suggest to replot them.

7 Line 263, when the door opened, whether the CO2 concentration would drop down? Drop down by how many percentages?

8 Lines 270-271, much more people should take the tram during the working days when compared to those in weekend, why the CO2 concentration in working days was only slightly higher than that in weekend?

Author Response

Overall, this work is interesting and important, I only have a few comments:

We appreciate Referee #1 for the time invested, great effort, and critical reading of the manuscript. We acknowledge the reviewer's comments and apologize for the mistakes.

We very much appreciate that Reviewer #1 finds this work interesting and important.

1 Lines 74-75, indeed, it is not that difficult to measure the aerosol directly by using the handheld instruments or simple sensors.

Agree. We appreciate this comment. We rewrote this paragraph including lines 76-79: “While there are handheld instruments or simple sensors to direct measure of aerosol concentrations, these instruments present different limitations such as they cannot discriminate human-exhaled versus environmental aerosols, usually they cover a limited range of particle diameter and hardly measure the submicronic particles.”

2 Please provide reference for this sentence “CO2 is co-expired with bioaerosols 77 that may contain SARS-CoV-2 in infected people.”

We value this observation. We have included the following references: [28][29][30]. Consequently, all references have been updated (line 82).

[28] Kappelt, N.; Russell, H.S.; Kwiatkowski, S.; Afshari, A.; Johnson, M.S. Correlation of Respiratory Aerosols and Metabolic Carbon Dioxide. Sustain. 2021, 13, 1–11, doi:10.3390/su132112203.

[29] Baselga, M.; Güemes, A.; Alba, J.J.; Schuhmacher, A.J. SARS-CoV-2 Droplet and Airborne Transmission Heterogeneity. J. Clin. Med. 2022, 11, 2607.

[30] Schade, W.; Reimer, V.; Seipenbusch, M.; Willer, U. Experimental Investigation of Aerosol and Co2 Dispersion for Evaluation of Covid-19 Infection Risk in a Concert Hall. Int. J. Environ. Res. Public Health 2021, 18, 1–11, doi:10.3390/ijerph18063037.

3 Figure 1, whether there exists any vent at the top of the Tram?

We acknowledge this observation. Dumpers where located close to the HVAC. Figure 1 has been updated including Dumpers in the Scheme.

4 In the tram, whether the air is fully well mixed?

As depicted in Figure 6 (Section 3.2), the variability between meters measurements is highly homogeneous. Although accumulation tendencies are typically observed in the central area of the Tram, it could be explained by the passenger distribution along the Tram. We can add more explanations in the manuscript if not completely clear.

5 Lines 208-210, the authors need to mark out the location of the meteorological station in Figure 2, if the meteorological station is far away from the tram station, I do not think the data can represent the ventilation capacity.

Thank you for the suggestion. The location of the Station has been included now in Figure 2.

6 All the figures presented in this work are not that clear, especially Figure 4 and Figure 6, suggest to replot them.

We updated Figures 4 and 6. We hope there are clearer now. We also added more detailed captions in all Figures as highlighted.

7 Line 263, when the door opened, whether the CO2 concentration would drop down? Drop down by how many percentages?

It is not possible to accurately determine the CO2 drop during door opening since the meters can measure every 1 minute and door opening lasts about 15 seconds. We do not know if the subsequent data is significant because the entry of new passengers on the tram can alter the measurement.

8 Lines 270-271, much more people should take the tram during the working days when compared to those in weekend, why the CO2 concentration in working days was only slightly higher than that in weekend?

In Zaragoza (Spain), the Tram is a very common means of transport and is widely used during the weekend. Although time patterns change (e.g. there are fewer passengers at 7 am), it is very normal to see crowded trams on Friday and Saturday afternoons. Also keep in mind that the frequency of the Tram is less during the weekend, so that more people accumulate inside the transport.

We reiterate our gratitude for Referee #1´s contributions; they positively improve and strengthened the manuscript.

Reviewer 2 Report

For the word file.

Author Response

We appreciate Reviewer #2 for the time invested, great effort, and critical reading of the manuscript. We acknowledge the reviewer's comments and apologize for the mistakes.

320-322 From the data in Figure 8 alone, it may not be possible to understand the relationship with weather data. Be clear about what you are comparing and expressing and how you are expressing it.

Thank you for your suggestion. We added key information into the Figure 8 for a better understanding of the the data representation. We also added lines 315-317 to improve objective compression.

445-447 It is natural that ventilation will be better if the outside air intake is maximized. As a result of the research, it is expected that minimum ventilation, such as how long to open the windows and which windows in the car should be opened and how, whether the minimum window opening has the maximum ventilation effect in the car.

We appreciate this observation. The tram units studied only have 2 small windows in the head and tail parts of the unit. While these can be opened in an emergency situation as Covid-19 pandemic, these windows should be closed according to the Spanish regulations (to avoid particles, dust, pollen etc). The opening of these windows also have an effect in the HVAC air renovations .

468-470 Evaluate weather conditions, please add some from the 307-312 results. These results due to be novel, original, and noteworthy results in this study. Any other results, conclusions, or recommendations should also be moved to the Discussion section.

Agree. We appreciate this comment. We included the following explanation in lines 486-489: “For example, in our study the weather substantially affected the ventilation ratio inside the Tram. On the days with the greatest wind, ppm/person rates of up to 44% lower were recorded with respect to the days with the least wind.” We adapted the conclusions and hope is clearer now.

By the way, the discussion section for 406-411 is also an interesting result. Can’t you explain it as a result and recommendation?

Agree. We added the following text (lines 472-480): “Additionaly, we propose to use the ppm/person ratio as the main indicator to comparise the exhaled CO2 measuerements on different scenarios. This ratio can be easily calculated by dividing the increase by the number of people. For example, if the increase in CO2 is 500 and there are 50 people, the ratio will be 10 ppm/person. In case of studying two separate days, for example with different weather, we can find that one day the ratio is 10 ppm/person and another day it is 30 ppm/person. With this information we can determine how the change of variables affects independently of the occupation”

486         You need to specify the cited reference

Agree. We appreciate this observation. We added “(by using Aireamos Covid Risk Airborne tool [31]; see 3.7 Section) (line 501). The tool we cite consists of a mathematical estimate of the risk of contagion based mainly on the level of CO2 inside. It has been developed within an international consortium in which some of the researchers involved in this article participate. So, based on the considerations in section 3.7, we can conclude that the results of this work do not imply a high risk of infection by aerosols.

490-491      Isn't the use of a hep-filter effective in removing stray virus particles? .

Agree. HEPA or similar high-retention filtration is very efficient at removing viral particles from the air. However, given its high pressure drop, it is not possible to implement it in the HVAC system of the air conditioning of the Zaragoza Tram.

491-495      As a research result, it has very little meaning. It is also of little significance to list it as a recommendation

Agree. We removed this paragraph from the main text. We also rephrased the last paragraph (lines 500-509).

We reiterate our gratitude for Referee #2´s contributions; they positively improve and strengthened the manuscript.
